# ADVERSARIAL GUIDED DIFFUSION MODELS FOR ADVERSARIAL PURIFICATION

## ABSTRACT

Diffusion model (DM) based adversarial purification (AP) has proven to be a powerful defense method that can remove adversarial perturbations and generate a purified example without threats. In principle, the pre-trained DMs can only ensure that purified examples conform to the same distribution of the training data, but it may inadvertently compromise the semantic information of input examples, leading to misclassification of purified examples. Recent advancements introduce guided diffusion techniques to preserve semantic information while removing the perturbations. However, these guidances often rely on distance measures between purified examples and diffused examples, which can also preserve perturbations in purified examples. To further unleash the robustness power of DM-based AP, we propose an adversarial guided diffusion model (AGDM) by introducing a novel adversarial guidance that contains sufficient semantic information but does not explicitly involve adversarial perturbations. The guidance is modeled by an auxiliary neural network obtained with adversarial training, considering the distance in the latent representations rather than at the pixel-level values. Extensive experiments are conducted on CIFAR-10, CIFAR-100 and ImageNet to demonstrate that our method is effective for simultaneously maintaining semantic information and removing the adversarial perturbations. In addition, comprehensive comparisons show that our method significantly enhances the robustness of existing DM-based AP, with an average robust accuracy improved by up to 7.30% on CIFAR-10. The code will be available upon acceptance.

## 1 INTRODUCTION

Deep neural networks (DNNs) have been shown to be vulnerable to adversarial examples (Szegedy et al., 2014), leading to disastrous implications. Since then, numerous methods have been proposed to defend against adversarial examples. Notably, adversarial training (AT, Goodfellow et al., 2015; Madry et al., 2018a) typically aims to retrain DNNs by using adversarial examples, achieving robustness over seen types of adversarial attacks. However, the model trained by AT is almost incapable of defending unseen types of adversarial attacks (Laidlaw et al., 2021; Dolatabadi et al., 2022). Another class of defense methods is adversarial purification (AP, Yoon et al., 2021) typically based on pre-trained generative models, aiming to eliminate potential adversarial perturbations for both clean or adversarial examples before feeding them into the classifier. Unlike AT technique, AP operates as a pre-processing step that can effectively defend against unseen types of attacks and does not require retraining classifiers. Hence, AP has emerged as a promising defense method and proven to be a powerful alternative to AT (Shi et al., 2021; Nie et al., 2022).

Recently, diffusion models (DMs, Ho et al., 2020; Song et al., 2020) have gained significant attention for their ability to generate high-quality images through diffusing images with Gaussian noises in a forward process and then denoise images in a reverse process. Motivated by the great success of DMs, Yoon et al. (2021); Nie et al. (2022) has shown that the pre-trained DM can be leveraged for adversarial purification as well as the theoretical analysis (Xiao et al., 2023; Carlini et al., 2023; Bai et al., 2024), which tries to purify either clean examples or adversarial examples by firstly adding Gaussian noises through the forward process with a number of timestep and then removing noises including adversarial perturbations to restore purified examples. Although DM-based AP can achieve remarkable robust performance and generalization ability to unseen attacks, some studies

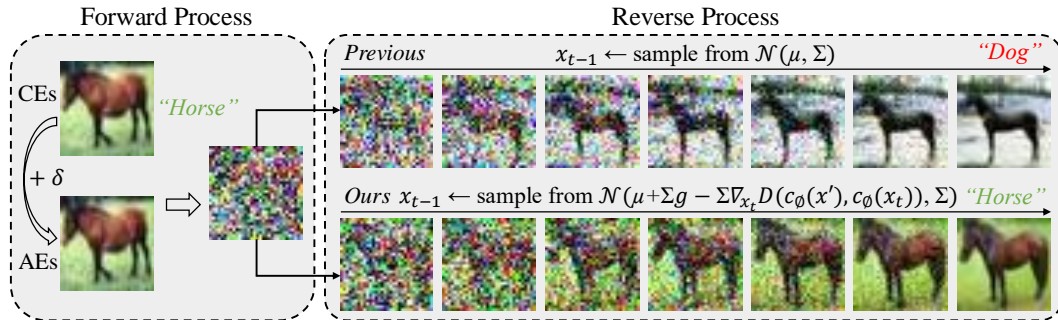

Figure 1: The scheme of diffusion-based purification. The clean examples (CEs) or adversarial examples (AEs) are firstly diffused with Gaussian noises and then removed the noise step by step. To make a clearer comparison, we set the step to 400. Unlike previous methods, our method can generate purified examples without changing its semantic information as well as groundtruth label.

(Wu et al., 2022; Wang et al., 2022) have shown that pre-trained DMs can restore the clean examples by removing adversarial perturbations, but they fail to ensure purified examples retain the same semantic information as original images. The reason is that adversarial perturbations can be gradually destroyed by Gaussian noises, but there is a risk that the semantic information of image might also be lost under too many timesteps in the forward process, leading to the purified examples being totally different from the expected clean examples. In principle, these DMs can only ensure that purified examples conform to the same distribution of the training data, but it may inadvertently compromise the semantic information of input examples, leading to misclassification of purified examples.

To address the above issue, one solution is to fine-tune the DMs using adversarial examples and their groundtruth labels by AToP (Lin et al., 2024), but this is computationally expensive. Another solution is to directly impose a guidance in the reverse process without fine-tuning the DMs. For instance, several guided diffusion techniques Wu et al. (2022); Wang et al. (2022); Bai et al. (2024) are introduced to preserve semantic information while removing the perturbations. The idea is to leverage the guidance to control the distribution of purified examples towards the distribution of input examples. However, these works often utilize the specific distance measures between purified examples and diffused examples as their guidance, yet the diffused examples inherently contain adversarial perturbation information. As a result, the adversarial perturbations cannot be removed completely, i.e., the perturbations are also preserved in the purified example, and thus the robustness performance is still unsatisfied (Kang et al., 2023; Chen et al., 2024). Consequently, the existing DM-based AP methods are still confronted with the formidable challenge of the trade-off between preserving semantic information and removing perturbations. This raises a critical question: *how to advance the robustness of DM-based AP against adversarial attacks while effectively removing perturbations and preserving semantic information?*

To further unleash the robustness power of DM-based AP, we propose an adversarial guided diffusion model (AGDM) by introducing a novel adversarial guidance during the reverse process, as illustrated in Figure 1. Unlike other guided diffusion models, we train an auxiliary neural network by adversarial training to model the probabilities of adversarial guidance that contains sufficient semantic information but does not explicitly involve adversarial perturbations. Furthermore, unlike AT optimizing the classifier and AToP optimizing the purifier, AGDM optimizes the guidance to better guide the diffusion-based purifier for adversarial purification, avoiding the huge computational burden issue of AToP. Finally, we heuristically create a conceptual diagram to review the whole process of DM-based AP and explain why AGDM can effectively remove the perturbations without sacrificing the semantic information, as shown in Figure 2. To demonstrate the effectiveness of our method, we empirically evaluate the performance by comparing with the latest AT and AP methods across various attacks, including AutoAttack (Croce & Hein, 2020), StAdv (Xiao et al., 2018), PGD (Madry et al., 2018b; Lee & Kim, 2023) and EOT (Athalye et al., 2018), on CIFAR-10, CIFAR-100 and ImageNet datasets under multiple classifier models. The results show that our method is effective for simultaneously maintaining semantic information and removing the adversarial perturbations, and exhibits robust generalization against unseen attacks. Specifically, on CIFAR-10, our method improves robust accuracy against AutoAttack by up to 8.26% compared to vanilla DMs. Furthermore, our results on

the robust evaluation of diffusion-based purification (Lee & Kim, 2023) manifest the necessity of adversarial guidance in diffusion models for AP, which improves robust accuracy by up to 9.53% compared to existing DM-based AP. Our contributions are summarized as follows.

- To further unleash the robustness power of DM-based AP, we propose an adversarial guided diffusion model (AGDM) by introducing adversarial guidance during the reverse process.
- The adversarial guidance is introduced and modeled by the probability of semantic representation, which can be learned by adversarial training an auxiliary neural network.
- We conduct extensive experiments to empirically evaluate our methods, which have demonstrated that the proposed method significantly improves the robustness power of DM-based AP, especially under the robust evaluation scheme.

## 2 PRELIMINARY

This section briefly reviews the adversarial training, adversarial purification, and diffusion models.

### 2.1 ADVERSARIAL TRAINING AND ADVERSARIAL PURIFICATION

Given a classifier $f_\gamma$ with input $x$ and output $y$, the adversarial attacks aim to find the adversarial examples $x'$ that can fool the classifier model $f_\gamma$. The adversarial examples can be obtained by

$$x' = x + \delta, \quad \delta = \arg\max_{\|\delta\| \leq \varepsilon} \mathcal{L}(f_\gamma(x + \delta), y),$$

where $\delta$ is an imperceptible adversarial perturbation and $\varepsilon$ is the maximum scale of perturbation. To defend against adversarial attacks, the most popular technique is adversarial training (AT, Goodfellow et al., 2015; Madry et al., 2018a), which requires the classifier $f_\gamma$ trained with adversarial examples by solving the min-max optimization problem, i.e., $\min_\gamma \mathbb{E}_{p_{\text{data}}(x,y)}[\max_{\|\delta\| \leq \varepsilon} \mathcal{L}(f_\gamma(x + \delta), y)]$.

Another technique is adversarial purification (AP, Yang et al., 2019), which aims to utilize a model $g_\theta$ that can purify adversarial examples before feeding them into the classifier $f_\gamma$, resulting in the same classification output with the clean example $x$, i.e., $f_\gamma(g_\theta(x + \delta)) = f_\gamma(x)$. It should be noted that the purifier model $g_\theta$ is not necessary to satisfy $g_\theta(x + \delta) = x$. As a plug-and-play module, $g_\theta$ is thus often achieved by a pre-trained generative model and can be integrated with any classifiers.

### 2.2 DIFFUSION MODELS

Diffusion models (Ho et al., 2020; Song et al., 2020) have proven to be a potent class of generative model capable of generating high-quality images through two distinct processes: a forward process transforming an image entirely into noise by gradually adding Gaussian noise, and a reverse process transforming noise into the generated image by gradually denoising image.

As described in DDPM (Ho et al., 2020), given a data distribution $x_0 \sim q(x_0)$, the forward process involves $T$ steps and any step $t$ can be rewritten as one direct sample from $q(x_t \mid x_0) = \mathcal{N}(x_t; \sqrt{\bar{\alpha}_t}x_0, (1 - \bar{\alpha}_t)\mathbf{I})$ where $\bar{\alpha}_t := \prod_{s=1}^{t} \alpha_s$ and $\alpha_s$ is a hyperparameter. The reverse process aims to restore the distribution $x_0$ from the Gaussian noise $x_T \sim \mathcal{N}(0, \mathbf{I})$ step by step using a U-Net $\epsilon_\theta$ trained by optimizing the loss function

$$L(\theta) = \mathbb{E}_{t,x_0,\epsilon}[\|\epsilon - \epsilon_\theta(\sqrt{\bar{\alpha}_t}x_0 + \sqrt{1 - \bar{\alpha}_t}\epsilon, t)\|^2],$$

where $\epsilon \sim \mathcal{N}(0, \mathbf{I})$ is an arbitrary Gaussian noise. Unlike the forward process that can be sampled directly in closed form, the reverse process requires $T$ steps to obtain $x_0$ from $x_T$. Therefore, as compared to other generative models, diffusion models are much slower in general.

## 3 METHODS

In this section, we first discuss the rationale for the necessity of adversarial guidance. Then, we propose a novel adversarial guided diffusion model, which can effectively remove adversarial perturbations without sacrificing semantic information and thus defend against various attacks. Finally, we provide our algorithm of the whole AP process for generating purified examples.

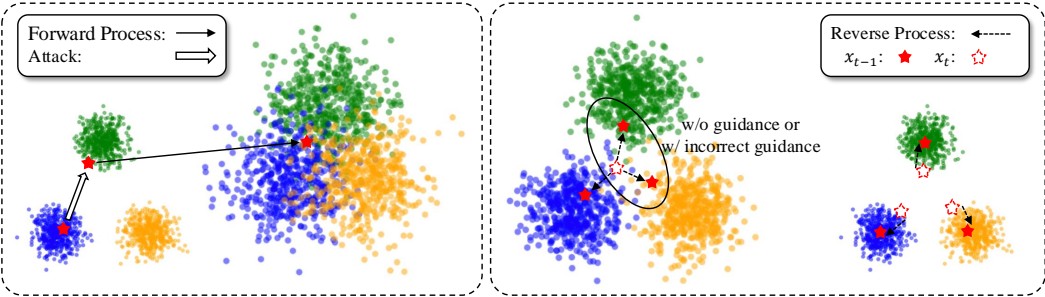

Figure 2: Overview of the forward process and reverse process. Different colored dots represent the data distributions of various categories. In the presence of attacks, without guidance or with improper guidance, the red star may move to the wrong category, thereby reducing robust accuracy.

### 3.1 MOTIVATION TO INTRODUCE ADVERSARIAL GUIDANCE

Based on Wang et al. (2022), we heuristically create a conceptual diagram as shown in Figure 2. The movement of the red star throughout the diagram illustrates the whole process. Initially, adversarial attacks shift ($\Longrightarrow$) the red star to a different data distribution, causing misclassification. Then, the data distribution is diffused ($\longrightarrow$) with the continuous addition of Gaussian noise. Finally, both Gaussian noise and adversarial perturbations are removed step by step during the reverse process, allowing the red star to gradually move back ($\leftarrow\!-$) to the clean data distribution. We heuristically argue that the limitations of existing DM-based AP methods stem from the lack of guidance or improper guidance, which may lead the red star into the clean data distribution but with the wrong category.

In Wang et al. (2022), a guidance of minimizing the distance (mean squared error) between adversarial examples and purified examples was introduced under the assumption that the distributions of adversarial examples and clean examples are close. This guidance can effectively prevent DM from generating a totally different image when diffused step $T$ is larger. However, it is also possible that purified examples will be too close to adversarial examples such that even adversarial perturbation is preserved. In Bai et al. (2024), an improved guidance was introduced, which uses contrastive loss for encouraging purified examples $x_{t-1}$ to be similar to $x_t$. However, this guidance also considered the distance between purified examples and diffused adversarial examples in terms of pixel values, thus the perturbations can be partially preserved. Our conjecture is that the distribution of purified examples should not be similar to that of adversarial examples, and the guidance should not use adversarial perturbations explicitly. To this end, we consider that the guidance can be designed over robust representations of purified examples and adversarial examples. Even if the distribution of their representation is similar, the purified examples can be ensured not to contain adversarial perturbations explicitly while also keeping the semantic information similar.

### 3.2 ADVERSARIAL GUIDED DIFFUSION MODEL

Unlike the previous guided diffusion models (Wu et al., 2022; Wang et al., 2022; Zhang et al., 2024; Bai et al., 2024), we are the first to introduce an adversarial guidance to the reverse process. Based on Dhariwal & Nichol (2021), a conditional distribution of purified example $x_t$ can be adopted as the reverse process, which is

$$p_{\theta,\phi}(x_{t-1} \mid x_t, y, x') \propto p_\theta(x_{t-1} \mid x_t) p_\phi(x' \mid x_t) p_\phi(y \mid x_t). \tag{1}$$

Note that our guidance not only contains adversarial examples $x'$ but also the predictive class probabilities $y$ to preserve semantic information. Based on the assumption that $y$ and $x'$ are conditionally independent given $x_t$, this conditional distribution can be factorized into the right terms in Eq. (1). Among them, $p_\theta(x_{t-1} \mid x_t)$ is the unconditional DDPM obtained by the pre-trained diffusion model. $p_\phi(x' \mid x_t)$ can be interpreted as the probability that $x_t$ will be eventually purified to a clean example having similar semantic information with $x'$, and $p_\phi(y \mid x_t)$ can be interpreted as the probability that $x_t$ will be purified to a clean example with predictive class probabilities close to $y$. Since we expect that the similarity of semantic information can be easily measured but without explicitly involving adversarial perturbations, an auxiliary neural network $c_\phi$ is introduced to map data into

latent representations that are convenient for classification. Therefore, to push the purified example $x_t$ close to $x'$ in terms of high-level representations rather than their pixel values, we adopt a heuristic probability approximation,

$$p_\phi(x' \mid x_t) \propto \exp(-\mathcal{D}(c_\phi(x'), c_\phi(x_t))), \tag{2}$$

where $\mathcal{D}(\cdot, \cdot)$ is the distance metric for measuring the similarity of the representations inferred by $c_\phi$ between $x'$ and $x_t$. In the reverse process, the purified example $x_t$ is encouraged to increase $p_\phi(x' \mid x_t)$, i.e., to decrease $\mathcal{D}(c_\phi(x'), c_\phi(x_t))$. This is the key technique which can avoid introducing the adversarial perturbations into the purified example due to the transformation by $c_\phi$ and the similarity of $(c_\phi(x'), c_\phi(x_t))$ does not necessarily lead to the similarity of $(x', x_t)$ in the pixel values. Due to that the label information is unavailable in the purification, we cannot compute $p_\phi(y \mid x_t)$ directly. Since we expect to encourage the class information of $x_{t-1}$ will not be changed dramatically by a reverse step, thus the predictive class probabilities can be approximated by $p_\phi(y \mid x_t) = \mathrm{softmax}(c_\phi(x_t))$.

Next, we further derive and explain how the above guidance can be leveraged in the reverse process of DM. For the first term in Eq. (1), we have

$$p_\theta(x_{t-1} \mid x_t) = \mathcal{N}(x_{t-1}; \mu_\theta(x_t, t), \sigma_t^2 \mathbf{I})$$

$$\log p_\theta(x_{t-1} \mid x_t) = -\frac{1}{2}(x_t - \mu)^\intercal \Sigma^{-1}(x_t - \mu) + C_1, \tag{3}$$

where $\mu := \mu_\theta(x_t, t)$ and $\Sigma := \sigma_t^2 \mathbf{I}$ are obtained by the pre-trained diffusion model and $C_1$ is a constant w.r.t. $x_t$. We omit the inputs of the functions $\mu, \Sigma$ for clarity, consistent with notations in Dhariwal & Nichol (2021). The second term in Eq. (1) can be approximated as

$$\log p_\phi(x' \mid x_t) \approx \log p_\phi(x' \mid x_t) \mid_{x_t=\mu} + (x_t - \mu)^\intercal \nabla_{x_t} \log p_\phi(x' \mid x_t) \mid_{x_t=\mu}$$

$$= (x_t - \mu)^\intercal \nabla_{x_t} \mathcal{D}(c_\phi(x'), c_\phi(x_t)) + C_2, \tag{4}$$

where $C_2$ is a constant w.r.t. $x_t$. Finally, for the last term in Eq. (1), we have

$$\log p_\phi(y \mid x_t) \approx \log p_\phi(y \mid x_t) \mid_{x_t=\mu} + (x_t - \mu)^\intercal \nabla_{x_t} \log p_\phi(y \mid x_t) \mid_{x_t=\mu}$$

$$= (x_t - \mu)^\intercal g + C_3, \tag{5}$$

where $g = \nabla_{x_t} \log p_\phi(y \mid x_t) = \nabla_{x_t} c_\phi(x_t)$ and $C_3$ is a constant w.r.t. $x_t$. By plugging Eqs. (3) to (5) into Eq. (1), we obtain the adjusted function with adversarial guidance,

$$\log p_{\theta,\phi}(x_{t-1} \mid x_t, y, x') = \log p(z) + C_4, \tag{6}$$

where $C_4$ is a constant and $z$ follows

$$z \sim \mathcal{N}(z; \mu + \Sigma g - \Sigma \nabla_{x_t} \mathcal{D}(c_\phi(x'), c_\phi(x_t)), \Sigma). \tag{7}$$

The full derivation is shown in Appendix A.1. The above derivation of guided sampling is valid for DDPM. It can also be extended to continuous-time diffusion models with details in Appendix A.2.

## 3.3 AGDM-BASED ADVERSARIAL PURIFICATION

Given an adversarial example $x'$, we first diffuse it by Gaussian noise with $t^*$ steps,

$$x_{t^*} = \sqrt{\bar{\alpha}_{t^*}} x' + \sqrt{1 - \bar{\alpha}_{t^*}} \epsilon, \quad \epsilon \sim \mathcal{N}(0, \mathbf{I}). \tag{8}$$

Then, in our robust reverse process, we can obtain the purified example $x_0$ by sampling $x_{t-1}$ from Eq. (7) with $t^*$ steps. Note that we add a scale $s$ to adjust the guidance, which can be regarded as a temperature (Kingma & Dhariwal, 2018) in the distribution, i.e., $p_\phi(x' \mid x_t)^s p_\phi(y \mid x_t)^s$. However, training this noise-conditioned guidance is challenging. For practical usage, we adopt the approximation $p_\phi(x' \mid x)^s p_\phi(y \mid x)^s$,

**Algorithm 1** AGDM-based AP, given diffusion model $(\mu_\theta(x_t, t), \sigma_t^2 \mathbf{I})$, auxiliary NN $c_\phi$, scale $s$.

---

**Input:** Adversarial example $x'$ and timestep $t^*$.
1: $x_{t^*} \leftarrow$ sample from Eq. (8)
2: **for** $t$ from $t^*$ to 1 **do**
3: $\quad \mu, \Sigma \leftarrow \mu_\theta(x_t, t), \sigma_t^2 \mathbf{I}$
4: $\quad$ # Vanilla $x_{t-1} \leftarrow$ sample from $\mathcal{N}(\mu, \Sigma)$
5: $\quad x_{t-1} \leftarrow$ sample from
$\quad \mathcal{N}(\mu + s\Sigma g - s\Sigma \nabla_{x_t} \mathcal{D}(c_\phi(x'), c_\phi(x_t)), \Sigma)$
6: **end for**
7: **return** Purified example $x_0$

where the guidance is trained on clean example $x$, as we will describe in the following paragraph. While this approximation works well in our experimental settings, an interesting future direction

would be theoretical justification of this approximated guidance as in Chung et al. (2022). Finally, the whole process of AGDM-based adversarial purification is presented in Algorithm 1.

To train the auxiliary neural network $c_\phi$, we utilize TRADES technique, which incorporates classification loss and discrepancy loss, i.e., $\min_\phi \mathbb{E}_{p_{\text{data}}(x,\overline{y})}[\mathcal{L}(c_\phi(x), \overline{y}) + \lambda \max_{\|\delta\| \leq \varepsilon} \mathcal{D}(c_\phi(x), c_\phi(x'))]$ (Zhang et al., 2019), where $x, \overline{y}$ are clean example and its groundtruth label, and $\lambda$ is a weighting hyperparameter. The discrepancy loss $\mathcal{D}(c_\phi(x), c_\phi(x'))$ is introduced to avoid the recovery of perturbation information while the classification loss $\mathcal{L}(c_\phi(x), \overline{y})$ is introduced to better preserve semantic information. Note that $c_\phi$ does not require to be a robust classifier, but having robust representations when facing adversarial perturbations.

## 4 RELATED WORKS

**Adversarial robustness:** To defend against adversarial attacks, researchers have developed various techniques aimed at enhancing the robustness of DNNs. Specifically, Zhang et al. (2019) propose TRADES that incorporates classification loss and discrepancy loss into adversarial training to enhance the robustness of classifiers. Lin et al. (2024) propose AToP that fine-tunes the generator-based purifier with adversarial training and makes it more suitable for robust classification tasks. Unlike Zhang et al. (2019) optimizing the classifier and Lin et al. (2024) optimizing the purifier, our method utilizes TRADES loss to train an auxiliary neural network to better guide diffusion model for adversarial purification, avoiding the substantial computational cost of adversarial training on DMs and effectively defending against unseen attacks.

**Diffusion model based adversarial purification:** Motivated by the great success of DMs, Yoon et al. (2021); Nie et al. (2022) utilize a pre-trained DM for adversarial purification and achieve remarkable performance in robust classification. In subsequent research, Wu et al. (2022); Wang et al. (2022) aim to further preserve semantic information by minimizing the distance between adversarial examples and purified examples. Bai et al. (2024) propose an improved guidance, which uses contrastive loss to encourage the purified examples from adjacent steps to be similar. However, both guidances utilize the distance measures in terms of pixel values, thus the perturbations can be partially preserved. Distinguishing with these methods, we leverage distance measures within latent representations from an auxiliary neural network rather than relying on pixel-level differences, avoiding the recovery of perturbation information. Additionally, Zhang et al. (2024) propose classifier guidance, which preserves semantic information by directly using the confidence score from the downstream classifier trained on clean examples. However, if attackers gain access to the classifier information used in the guidance, it may lead to incorrect confidence under adversarial attacks. In contrast, we train the auxiliary neural network using adversarial training to provide more robust guidance.

## 5 EXPERIMENTS

In this section, we conduct extensive experiments on CIFAR-10, CIFAR-100 and ImageNet across various classifier models on attack benchmarks. Compared with the AT and AP methods, our method achieves state-of-the-art robustness and exhibits generalization ability against unseen attacks. Furthermore, we undertake a more comprehensive evaluation against more powerful attacks. The results show that our method can significantly improve the performance of the DM-based AP.

### 5.1 EXPERIMENTAL SETUP

**Datasets and classifiers:** We conduct extensive experiments on CIFAR-10, CIFAR-100 (Krizhevsky et al., 2009) and ImageNet (Deng et al., 2009) to empirically validate the effectiveness of the proposed methods against adversarial attacks. For the classifier models, we utilize the pre-trained ResNet (He et al., 2016) and WideResNet (Zagoruyko & Komodakis, 2016).

**Adversarial attacks:** We evaluate our method against AutoAttack (Croce & Hein, 2020) as one benchmark, which is a common attack that combines both white-box and black-box attacks. To consider unseen attacks without $l_p$-norm, we utilize spatially transformed adversarial examples (StAdv, Xiao et al., 2018) for evaluation. Additionally, following the guidance of Lee & Kim (2023), we utilize projected gradient descent (PGD, Madry et al., 2018b) with expectation over time (EOT, Athalye et al., 2018) for a more comprehensive evaluation of the diffusion-based purification.

**Evaluation metrics:** We evaluate the performance of defense methods using two metrics: standard accuracy and robust accuracy, obtained by testing on clean examples and adversarial examples, respectively. Due to the high computational cost of testing models with multiple attacks, following guidance by Nie et al. (2022), we randomly select 512 images from the test set for robust evaluation.

**Training details:** According to Zhang et al. (2019); Dhariwal & Nichol (2021) and experiments, we set the diffusion timestep $t^* = 70$, the scale $s = 1.0$ and the weighting scale $\lambda = 6.0$. Unless otherwise specified, all experiments presented in the paper are conducted under these hyperparameters and done using the NVIDIA RTX A5000 with 24GB GPU memory and CUDA v11.7 in PyTorch v1.13.1 (Paszke et al., 2019).

## 5.2 COMPARISON WITH THE STATE-OF-THE-ART METHODS

We evaluate our method of defending against AutoAttack $l_\infty$ and $l_2$ threat models (Croce & Hein, 2020) and compare with the state-of-the-art methods as listed in RobustBench (Croce et al., 2021).

Table 1: Standard and robust accuracy against AutoAttack $l_\infty$ threat ($\epsilon = 8/255$) on CIFAR-10. ($^\dagger$the methods use additional synthetic images.)

| Defense method | Extra data | Standard Acc. | Robust Acc. |
|---|---|---|---|
| Zhang et al. (2020) | ✓ | 85.36 | 59.96 |
| Gowal et al. (2020) | ✓ | 89.48 | 62.70 |
| Bai et al. (2023) | ✓$^\dagger$ | 95.23 | 68.06 |
| Gowal et al. (2021) | ×$^\dagger$ | 88.74 | 66.11 |
| Wang et al. (2023) | ×$^\dagger$ | 93.25 | 70.69 |
| Peng et al. (2023) | ×$^\dagger$ | 93.27 | 71.07 |
| Rebuffi et al. (2021) | × | 87.33 | 61.72 |
| Wang et al. (2022) | × | 84.85 | 71.18 |
| Lin et al. (2024) | × | 90.62 | 72.85 |
| Ours | × | 90.82 | **78.12** |

Table 2: Standard and robust accuracy against AutoAttack $l_2$ threat ($\epsilon = 0.5$) on CIFAR-10.

| Defense method | Extra data | Standard Acc. | Robust Acc. |
|---|---|---|---|
| Augustin et al. (2020) | ✓ | 92.23 | 77.93 |
| Gowal et al. (2020) | ✓ | 94.74 | 80.53 |
| Wang et al. (2023) | ×$^\dagger$ | 95.16 | 83.68 |
| Ding et al. (2019) | × | 88.02 | 67.77 |
| Rebuffi et al. (2021) | × | 91.79 | 78.32 |
| Zhang et al. (2024) | × | 92.58 | 83.13 |
| Bai et al. (2024) | × | 93.75 | 84.38 |
| Ours | × | 90.82 | **86.84** |

Table 3: Standard and robust accuracy against AutoAttack $l_\infty$ ($\epsilon = 8/255$) on CIFAR-100.

| Defense method | Extra data | Standard Acc. | Robust Acc. |
|---|---|---|---|
| Hendrycks et al. (2019) | ✓ | 59.23 | 28.42 |
| Debenedetti et al. (2023) | ✓ | 70.76 | 35.08 |
| Cui et al. (2023) | ×$^\dagger$ | 73.85 | 39.18 |
| Wang et al. (2023) | ×$^\dagger$ | 75.22 | 42.67 |
| Pang et al. (2022) | × | 63.66 | 31.08 |
| Jia et al. (2022) | × | 67.31 | 31.91 |
| Cui et al. (2023) | × | 65.93 | 32.52 |
| Ours | × | 69.73 | **46.09** |

Table 4: Robust accuracy against AutoAttack $l_\infty$ threat ($\epsilon = 8/255$) and $l_2$ threat ($\epsilon = 0.5$). ([1]the method without guidance, [2]the method with guidance, [3]the method with adversarial guidance.)

| Defense method | CIFAR 10, $l_\infty$ | CIFAR 10, $l_2$ | CIFAR 100, $l_\infty$ |
|---|---|---|---|
| Nie et al. (2022) [1] | 70.64 | 78.58 | 42.19 |
| Zhang et al. (2024) [2] | 73.05 | 83.13 | 40.62 |
| Ours [3] | **78.12** | **86.84** | **46.09** |

**Result analysis on AutoAttack:** Tables 1 to 3 show the performance of various defense methods against AutoAttack $l_\infty$ ($\epsilon = 8/255$) and $l_2$ ($\epsilon = 0.5$) threats on CIFAR-10 and CIFAR-100 datasets using WideResNet-28-10. Our method outperforms all other methods without extra data (the dataset introduced by Carmon et al. (2019)) and additional synthetic data in terms of both standard accuracy and robust accuracy against $l_\infty$ threat. Specifically, as compared to the second-best method, our method improves the robust accuracy by 5.27% on CIFAR-10 and by 13.57% on CIFAR-100. Under the $l_2$ threat on CIFAR-10, our method outperforms all methods in terms of robust accuracy with an improvement of 2.46% over the second-best guided DM-based AP method Bai et al. (2024). These results are consistent across datasets and threats, confirming the effectiveness of our method for adversarial purification and its potential as a powerful defense technique.

**Comparison analysis on guidance:** Table 4 shows the comparative robust accuracy of three different pipelines, including the method without guidance (Nie et al., 2022), the method with guidance (Zhang et al., 2024), and our method with adversarial guidance. We can see that within the existing guidance,

Table 5: Standard accuracy and robust accuracy against AutoAttack $l_\infty$ ($\epsilon = 8/255$), $l_2$ ($\epsilon = 1$) and StAdv non-$l_p$ ($\epsilon = 0.05$) threat models on CIFAR-10 with ResNet-50 model. We keep the same settings with Nie et al. (2022), where the diffusion timestep $t^* = 125$.

| Defense method | Standard Acc. | AA $l_\infty$ | AA $l_2$ | StAdv |
|---|---|---|---|---|
| Standard Training | 94.8 | 0.0 | 0.0 | 0.0 |
| Adv. Training with $l_\infty$ (Laidlaw et al., 2021) | 86.8 | 49.0 | 19.2 | 4.8 |
| Adv. Training with $l_2$ (Laidlaw et al., 2021) | 85.0 | 39.5 | 47.8 | 7.8 |
| Adv. Training with StAdv (Laidlaw et al., 2021) | 86.2 | 0.1 | 0.2 | 53.9 |
| Adv. Training with all (Laidlaw et al., 2021) | 84.0 | 25.7 | 30.5 | 40.0 |
| PAT-self (Laidlaw et al., 2021) | 82.4 | 30.2 | 34.9 | 46.4 |
| Adv. CRAIG (Dolatabadi et al., 2022) | 83.2 | 40.0 | 33.9 | 49.6 |
| DiffPure (Nie et al., 2022) | 88.2 | 70.0 | 70.9 | 55.0 |
| AToP (Lin et al., 2024) | 89.1 | 71.2 | 73.4 | 56.4 |
| AGDM (Ours) | 89.3 | **78.1** | **79.6** | **59.4** |

DM-based AP has better robustness on CIFAR-10, but on more complex tasks, the robustness on CIFAR-100 actually decreases slightly. In contrast, our method consistently outperforms under all situations. Specifically, our method improves the robust accuracy by 5.07% against AutoAttack $l_\infty$, and by 3.71% against AutoAttack $l_2$ on CIFAR-10, respectively. Furthermore, it shows an improvement of 3.90% on CIFAR-100. This substantial improvement can be attributed to the targeted refinement that introduces adversarial guidance during the reverse process, effectively removing the perturbations without sacrificing the semantic information of purified examples. Unlike existing guided DM-based AP that may preserve a portion of perturbations, our AGDM-based AP prioritizes modifications that are beneficial for robust classification. This is also validated in Table 6.

## 5.3 DEFEND AGAINST UNSEEN ATTACKS

As previously mentioned, unlike AT, AP can defend against unseen attacks, which is an important metric for evaluating AP. To demonstrate the generalization ability of AGDM, we conduct experiments under several attacks with varying constraints (AutoAttack $l_\infty$, $l_2$ and StAdv non-$l_p$ threat models) on CIFAR-10 with ResNet-50. Table 5 shows that AT methods (PAT, CRAIG) are limited in defending

Table 6: Standard and robust accuracy against PGD+EOT (left: $l_\infty$, $\epsilon = 8/255$; right: $l_2$, $\epsilon = 0.5$) on CIFAR-10. We keep the same settings with Lee & Kim (2023), the diffusion timestep $t^* = 100$. ([1]the method without guidance, [2]the method with guidance, [3]the method with adversarial guidance.)

| Type | Defense method | Standard Acc. | Robust Acc. | Type | Defense method | Standard Acc. | Robust Acc. |
|---|---|---|---|---|---|---|---|
| WideRestNet-28-10 | | | | WideRestNet-28-10 | | | |
| AT | Pang et al. (2022) | 88.62 | 64.95 | AT | Sehwag et al. (2021) | 90.93 | 83.75 |
| | Gowal et al. (2020) | 88.54 | 65.93 | | Rebuffi et al. (2021) | 91.79 | 85.05 |
| | Gowal et al. (2021) | 87.51 | 66.01 | | Augustin et al. (2020) | 93.96 | 86.14 |
| AP | Wang et al. (2022) [2] | 93.50 | 24.06 | AP | Wang et al. (2022) [2] | 93.50 | - |
| | Yoon et al. (2021) | 85.66 | 33.48 | | Yoon et al. (2021) | 85.66 | 73.32 |
| | Nie et al. (2022) [1] | 91.41 | 46.84 | | Nie et al. (2022) [1] | 91.41 | 79.45 |
| | Lee & Kim (2023) | 90.16 | 55.82 | | Lee & Kim (2023) | 90.16 | 83.59 |
| | Ours [3] | 90.42 | **64.06** | | Ours [3] | 90.42 | **85.55** |
| WideRestNet-70-16 | | | | WideRestNet-70-16 | | | |
| AT | Gowal et al. (2020) | 91.10 | 68.66 | AT | Rebuffi et al. (2021) | 92.41 | 86.24 |
| | Gowal et al. (2021) | 88.75 | 69.03 | | Gowal et al. (2020) | 94.74 | 88.18 |
| | Rebuffi et al. (2021) | 92.22 | 69.97 | | Rebuffi et al. (2021) | 95.74 | 89.62 |
| AP | Yoon et al. (2021) | 86.76 | 37.11 | AP | Yoon et al. (2021) | 86.76 | 75.66 |
| | Nie et al. (2022) | 92.15 | 51.13 | | Nie et al. (2022) | 92.15 | 82.97 |
| | Lee & Kim (2023) | 90.53 | 56.88 | | Lee & Kim (2023) | 90.53 | 83.75 |
| | Ours | 90.43 | **66.41** | | Ours | 90.43 | **85.94** |

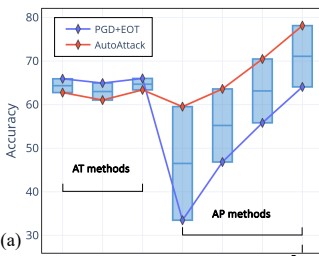 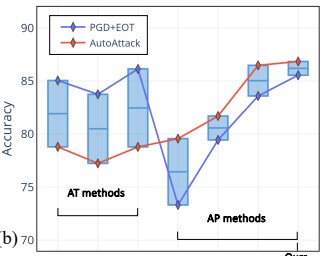 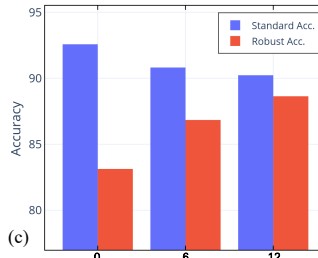

Figure 3: Comparison of robust accuracy against PGD+EOT and AutoAttack with (a) $l_\infty$ ($\epsilon = 8/255$) threat model and (b) $l_2$ ($\epsilon = 0.5$) threat model on CIFAR-10 with WideResNet-28-10. The line in the middle of the box represents the average robust accuracy of two attacks. (c) Accurcy-robustness trade-off against $l_2$ ($\epsilon = 0.5$) threat model discussed in Appendix B.

against unseen attacks and can only defend against known attacks (as indicated by the accuracy with an underscore) that they are trained with. In contrast, AP methods (DiffPure, AToP, AGDM) exhibit great generalization, defending against unseen attacks without significantly decreasing the standard accuracy, which is also validated in Table 8 in the Appendix. Specifically, compared to the best AT method, AGDM improved standard accuracy by 6.1%, and compared to the second-best AP method, it improved average robust accuracy by 5.4%.

## 5.4 ROBUST EVALUATION OF DIFFUSION-BASED PURIFICATION

Recently, Lee & Kim (2023); Chen et al. (2024) conducted a thorough investigation into the evaluation of DM-based AP, proposing a robust evaluation guideline using PGD+EOT. To undertake a more comprehensive evaluation, we further evaluate our method following the guidelines in this subsection.

Table 7: Standard accuracy and robust accuracy against PGD+EOT $l_\infty$ ($\epsilon = 4/255$) on ImageNet with ResNet-50. The diffusion timestep $t^* = 75$.

| Type | Defense method | Standard Acc. | Robust Acc. |
|------|---------------|:------:|:------:|
| AT | Wong et al. (2019) | 53.83 | 28.04 |
| | Engstrom et al. (2019) | 62.42 | 33.20 |
| | Salman et al. (2020) | 63.86 | 39.11 |
| AP | Nie et al. (2022) | 71.48 | 38.71 |
| | Lee & Kim (2023) | 70.74 | 42.15 |
| | Ours | 68.75 | **45.90** |

**Result analysis on PGD+EOT:** Initially, due to the substantial memory requirements needed to compute the direct gradient of the full defense process, most previous DM-based AP methods have not yet been evaluated using PGD+EOT. Recent works optimize the attack process and evaluate DM-based AP methods more comprehensively, revealing their vulnerability to PGD+EOT. As shown in Table 6, DiffPure (Nie et al., 2022) shows robust accuracy of 46.84%, significantly lower than the reported robust accuracy of 70.64% with AutoAttack. This large discrepancy again raises doubts about the robustness of DM-based AP methods. In contrast, our method achieves robust accuracy of 64.06% against $l_\infty$ and 85.55% against $l_2$, as compared to the second-best method, our method improves the robust accuracy by 8.24% and by 1.96%, respectively. Table 7 shows the results on ImageNet, and the observations are basically consistent with CIFAR-10, supporting our method as a powerful defense technique and more effective than previous DM-based AP methods. Furthermore, Table 6 also presents the comparative results of three guidance pipelines of diffusion model, where our method improves the average of standard accuracy and robust accuracy by 8.12% and 18.46% compared to the other two guidance pipelines, respectively.

**Comparison analysis between PGD+EOT and AutoAttack:** Figure 3a and 3b show the comparison between PGD+EOT and AutoAttack on $l_\infty$ and $l_2$ threat models. Under different attacks, AT methods (Gowal et al., 2020; 2021; Pang et al., 2022) and AP methods (Yoon et al., 2021; Nie et al., 2022; Lee & Kim, 2023) exhibit significant differences in robust accuracy. AT performs better under PGD+EOT, while AP shows superior performance under AutoAttack. Typically, robustness evaluation is based on the worst-case results of the robust accuracy. Under this criterion, our method still outperforms all AT and AP methods. Furthermore, as compared to the second-best method on both attacks, our method improves the average robust accuracy by 6.39% against $l_\infty$ and 1.16% against $l_2$, respectively. Such a significant margin from different attacks highlights the robustness of our method, particularly in worst-case results of the robust accuracy across PGD+EOT and AutoAttack.

## 6 CONCLUSION

In this paper, we propose an adversarial guided diffusion model (AGDM) for adversarial purification, which can enhance the robustness power of DM-based AP by introducing adversarial guidance during the reverse process. We conduct extensive experiments to empirically demonstrate that AGDM is effective for simultaneously maintaining semantic information and removing the adversarial perturbations, and exhibits robust generalization against unseen attacks.

**Limitations and discussion:** Similar to previous studies (Nie et al., 2022; Wang et al., 2022; Zhang et al., 2024; Bai et al., 2024), our proposed AGDM also features a time-consuming reverse process. Additionally, this paper adopts a heuristic perspective, we aim to use theoretical analysis in the future to more comprehensively demonstrate the effectiveness of AGDM. In summary, we leave the study of utilizing our adversarial guidance in more reliable and fast sampling strategies for future research.

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

# A  PROOFS OF ADVERSARIAL GUIDED DIFFUSION MODEL (AGDM)

## A.1  ROBUST REVERSE PROCESS FOR DDPM

In the reverse process with adversarial guidance, similar to Dhariwal & Nichol (2021), we start by defining a conditional Markovian noising process $\hat{q}$ similar to $q$, and assume that $\hat{q}(y, x'|x_0)$ is an available label distribution and adversarial example (AE) for each image.

$$\hat{q}(x_0) := q(x_0)$$
$$\hat{q}(y, x'|x_0) := \text{Label and AE per image}$$
$$\hat{q}(x_{t+1}|x_t, y, x') := q(x_{t+1}|x_t) \tag{9}$$
$$\hat{q}(x_{1:T}|x_0, y, x') := \prod_{t=1}^{T} \hat{q}(x_t|x_{t-1}, y, x').$$

When $\hat{q}$ is not conditioned on $\{y, x'\}$, $\hat{q}$ behaves exactly like $q$,

$$\begin{aligned}
\hat{q}(x_{t+1}|x_t) &= \int_{y,x'} \hat{q}(x_{t+1}, y, x'|x_t) \, dy dx' \\
&= \int_{y,x'} \hat{q}(x_{t+1}|x_t, y, x') \hat{q}(y, x'|x_t) \, dy dx' \\
&= \int_{y,x'} q(x_{t+1}|x_t) \hat{q}(y, x'|x_t) \, dy dx' \\
&= q(x_{t+1}|x_t) \int_{y,x'} \hat{q}(y, x'|x_t) \, dy dx' \\
&= q(x_{t+1}|x_t) \\
&= \hat{q}(x_{t+1}|x_t, y, x').
\end{aligned} \tag{10}$$

Following similar logic, we have: $\hat{q}(x_{1:T}|x_0) = q(x_{1:T}|x_0)$ and $\hat{q}(x_t) = q(x_t)$. From the above derivation, it is evident that the conditioned forward process is identical to unconditioned forward process. According to Bayes rule, the reverse process $\hat{q}$ satisfies $\hat{q}(x_t|x_{t+1}) = q(x_t|x_{t+1})$.

$$\begin{aligned}
\hat{q}(y, x'|x_t, x_{t+1}) &= \frac{\hat{q}(x_{t+1}|x_t, y, x')\hat{q}(y, x'|x_t)}{\hat{q}(x_{t+1}|x_t)} \\
&= \hat{q}(y, x'|x_t).
\end{aligned} \tag{11}$$

For conditional reverse process $\hat{q}(x_t|x_{t+1}, y, x')$,

$$\begin{aligned}
\hat{q}(x_t|x_{t+1}, y, x') &= \frac{\hat{q}(x_t, x_{t+1}, y, x')}{\hat{q}(x_{t+1}, y, x')} \\
&= \frac{\hat{q}(x_t, x_{t+1}, y, x')}{\hat{q}(y, x'|x_{t+1})\hat{q}(x_{t+1})} \\
&= \frac{\hat{q}(x_t|x_{t+1})\hat{q}(y, x'|x_t, x_{t+1})\hat{q}(x_{t+1})}{\hat{q}(y, x'|x_{t+1})\hat{q}(x_{t+1})} \\
&= \frac{\hat{q}(x_t|x_{t+1})\hat{q}(y, x'|x_t, x_{t+1})}{\hat{q}(y, x'|x_{t+1})} \\
&= \frac{\hat{q}(x_t|x_{t+1})\hat{q}(y, x'|x_t)}{\hat{q}(y, x'|x_{t+1})} \\
&= \frac{q(x_t|x_{t+1})\hat{q}(y, x'|x_t)}{\hat{q}(y, x'|x_{t+1})}.
\end{aligned} \tag{12}$$

Here $\hat{q}(y, x'|x_{t+1})$ does not depend on $x_t$. Then, by assuming the label $y$ and adversarial example $x'$ are conditionally independent given $x_t$, we can set $\bar{t} = t + 1$ and rewrite the above equation as $\hat{q}(x_{\bar{t}-1}|x_{\bar{t}}, y, x') = Z \cdot q(x_{\bar{t}-1}|x_{\bar{t}})\hat{q}(x'|x_{\bar{t}})\hat{q}(y|x_{\bar{t}})$ where $Z$ is a constant.

### A.2 ROBUST REVERSE PROCESS FOR CONTINUOUS-TIME DIFFUSION MODELS

In the main text, we only showcased the preliminaries and the corresponding robust reverse process related to DDPM, but our method can also be extended to continuous-time diffusion models (Song et al., 2020). The continuous-time DMs build on the idea of DDPM, employ stochastic differential equations (SDE) to describe the diffusion process as follows,

$$dx = F(x,t)dt + G(t)dw, \tag{13}$$

where $w$ represents a standard Brownian motion, $F(x,t)$ represents the drift of $x_t$ and $G(t)$ represents the diffusion coefficient.

By starting from sample of Eq. 13 and revesing the process, Song et al. (2020) run backward in time and given by the reverse-time SDE,

$$dx = [F(x,t) - G(t)^2 \nabla_x \log p_t(x)]dt + G(t)d\bar{w}, \tag{14}$$

where $\bar{w}$ represents a standard reverse-time Brownian motion and $dt$ represents the infinitesimal time step. Similar to DDPM, the continuous-time diffusion model also requires training a network to estimate the time-dependent function $\nabla_x \log p_t(x)$. One common approach is to use a score-based model $s_\theta(x,t)$ (Song et al., 2020; Kingma et al., 2021). Subsequently, the reverse-time SDE can be solved by minimizing the score matching loss (Song & Ermon, 2019),

$$\mathcal{L}_\theta = \int_0^T \lambda(t)\mathbb{E}[\|s_\theta(x_t,t) - \nabla_{x_t} \log p_{0t}(x_t|x_0)\|^2]dt, \tag{15}$$

where $\lambda(t)$ is a weighting function, and $p_{0t}$ is the transition probability from $x_0$ to $x_t$, where $x_0 \sim p_0(x)$ and $x_t \sim p_{0t}(x_t|x_0)$.

In the robust reverse process of continuous-time DMs, similar to Song et al. (2020), we suppose the initial state distribution is $p_0(x(0) \mid y, x')$ based on Eq. 14. Subsequently, using Anderson (1982) for the reverse process, we have

$$dx = \left\{F(x,t) - \nabla \cdot \left[G(t)G(t)^T\right] - G(t)G(t)^T \nabla_x \log p_t(x|y,x')\right\} dt + G(t)d\bar{w}. \tag{16}$$

Given a diffusion process $x_t$ with SDE and score-based model $s_{\theta*}(x,t)$, we firest observe that

$$\nabla_x \log p_t(x_t|y,x') = \nabla_x \log \int p_t(x_t|y_t,y,x')p(y_t|y,x')dy_t, \tag{17}$$

where $y_t$ is defined via $x_t$ and the forward process $p(y_t \mid x_t)$. Following the two assumptions by Song et al. (2020): $p(y_t \mid y, x')$ is tractable; $p_t(x_t|y_t,y,x') \approx p_t(x_t|y_t)$, we have

$$\begin{aligned}
\nabla_x \log p_t(x_t|y,x') &\approx \nabla_x \log \int p_t(x_t|y_t)p(y_t|y,x')\,dydx' \\
&\approx \nabla_x \log p_t(x_t|\hat{y}_t) \\
&= \nabla_x \log p_t(x_t) + \nabla_x \log p_t(\hat{y}_t|x_t) \\
&\approx s_{\theta*}(x_t,t) + \nabla_x \log p_t(\hat{y}_t|x_t),
\end{aligned} \tag{18}$$

where $\hat{y}_t$ is a sample from $p(y_t|y,x')$. Then, by assuming the label $y$ and adversarial example $x'$ are conditionally independent given $x_t$, we can update Eq. 14 with above formula, and obtain a new denoising model $\bar{\epsilon}$ with the guidance of label $y$ and adversarial example $x'$,

$$dx_t = \left[F(x,t) - G^2(t)(\nabla_x \log p_t(x) + \nabla_x \log p_t(y|x) + \nabla_x \log p_t(x'|x))(x,t)\right] dt + G(t)\,d\bar{w}. \tag{19}$$

## B COMPARISON WITH AT, ATOP AND AGDM

To Enhance the existing pre-trained generator-based purification architecture to further improve robust accuracy against attacks. Lin et al. (2024) propose adversarial training on purification (AToP). Based on pre-trained model, they redesign the loss function to fine-tune the purifier model using adversarial loss.

Pre-training stage:

$$L_{\theta_g} = L_g(x, \theta_g). \tag{20}$$

Fine-tuning stage:

$$L_{\theta_g} = L_g(x', \theta_g) + s \cdot L_{cls}(x', \overline{y}, \theta_g, \theta_f) = L_g(x', \theta_g) + s \cdot \max_{\delta} CE\left\{\overline{y}, f(g(x', \theta_g))\right\}, \tag{21}$$

where $L_g$ represents the original generative loss function of the generator model, which trained on clean examples and generates images similar to clean examples. During fine-tuning, AToP input the adversarial examples $x'$ to optimize generator with generative loss, and further optimize the generator model with the adversarial loss $L_{cls}$, which is the cross-entropy loss between the output of $x'$ and the ground truth $\overline{y}$. However, training the generator with adversarial examples can lead to a decline in the performance on clean examples, thereby reducing standard accuracy. To address this issue, we utilize adversarial training (TRADES, Zhang et al., 2019) to train the neural network $c_\phi$ for adversarial guidance with classification loss on clean examples $x$ and discrepancy loss on adversarial examples $x'$ and clean examples $x$.

$$\min_\phi \mathbb{E}_{p_{\text{data}}(x,\overline{y})}[\underbrace{\mathcal{L}(c_\phi(x), \overline{y})}_{\text{for accuracy}} + \lambda \underbrace{\max_{\delta \leq \varepsilon} \mathcal{D}(c_\phi(x), c_\phi(x'))}_{\text{for robustness}}], \tag{22}$$

where $\lambda$ is a weighting scale to balance the accuracy-robustness trade-off. To facilitate clearer comparison, we have used the same notation as AToP to represent the TRADES loss function, which differs from the actual loss function.

$$L_{\theta_g} = L_g(x, \theta_g) + s_1 \cdot L_{cls}(x, \overline{y}, \theta_g, \theta_f) + s_2 \cdot L_{dis}(x, x', \theta_g, \theta_f)$$
$$= L_g(x, \theta_g) + s_1 \cdot CE\left\{\overline{y}, f(g(x, \theta_g))\right\} + s_2 \cdot KL\left\{f(g(x, \theta_g)), f(g(x', \theta_g))\right\}. \tag{23}$$

Distinct from Eq. 21, in Eq. 23 we revert the input of the first two terms back to the clean examples $x$. By increasing the weight of $s_1$, we can improve the standard accuracy on clean examples. Additionally, the new constraint term $L_{dis}$ is the KL divergence between the feature map from the clean example $x$ and the adversarial example $x'$. By increasing the weight of $s_2$, we can improve the robust accuracy on adversarial examples.

**Accuracy-robustness trade-off:** Figure 3c shows the performance against AutoAttack $l_2$ ($\epsilon = 0.5$) threat models on CIFAR-10 with different weighting scales $\lambda$. We observe that as the weighting scale $\lambda$ increasing, the robust accuracy increases while the standard accuracy decreases, which verifies Eq. (22) on the trade-off between robustness and accuracy. To our best knowledge, this is the first to discuss the accuracy-robustness trade-off challenge in pre-trained generator-based purification, which might be a significant contribution to advance the development of this field.

In summary, we follow AT and AToP, but the proposed AGDM is completely different from them. Fundamentally, AT optimizes the classifier, AToP optimizes the purifier, while AGDM optimizes a guidance to better guide the diffusion model in adversarial purification.

## C  ADDITIONAL EXPERIMENTS AND VISUALIZATION

Table 8: Standard accuracy and robust accuracy against PGD+EOT $l_\infty$ ($\epsilon = 8/255$), $l_2$ ($\epsilon = 0.5$) threat models on CIFAR-10.

| Method | Classifier | Standard Acc. | $l_\infty$ | $l_2$ | Avg. |
|---|---|---|---|---|---|
| Yoon et al. (2021) | WideRestNet-28-10 | 85.66 | 33.48 | 73.32 | 64.15 |
| Yoon et al. (2021) | WideRestNet-70-16 | 86.76 | 37.11 | 75.66 | 66.51 |
| Nie et al. (2022) | WideRestNet-28-10 | 91.41 | 46.84 | 79.45 | 72.57 |
| Nie et al. (2022) | WideRestNet-70-16 | 92.15 | 51.13 | 82.97 | 75.42 |
| Lee & Kim (2023) | WideRestNet-28-10 | 90.16 | 55.82 | 83.59 | 76.52 |
| Lee & Kim (2023) | WideRestNet-70-16 | 90.53 | 56.88 | 83.75 | 77.05 |
| AGDM (Ours) | WideRestNet-28-10 | 90.42 | 64.06 | 85.55 | 80.01 |
| AGDM (Ours) | WideRestNet-70-16 | 90.43 | **66.41** | **85.94** | **80.93** |

To undertake a more comprehensive evaluation as shown in Table 8, we further evaluate our method against PGD+EOT to show the robustness generalization of AGDM. Compared to the second-best

method, AGDM improves the average robust accuracy by 5.10% and 5.86% on WideRestNet-28-10 and WideRestNet-70-16, respectively. Although AGDM has a slightly lower standard accuracy compared to DiffPure, it achieves the best average accuracy including standard accuracy and robust accuracy, supporting our discussion in the main text that AGDM exhibits great generalization, defending against unseen attacks without significantly decreasing the standard accuracy

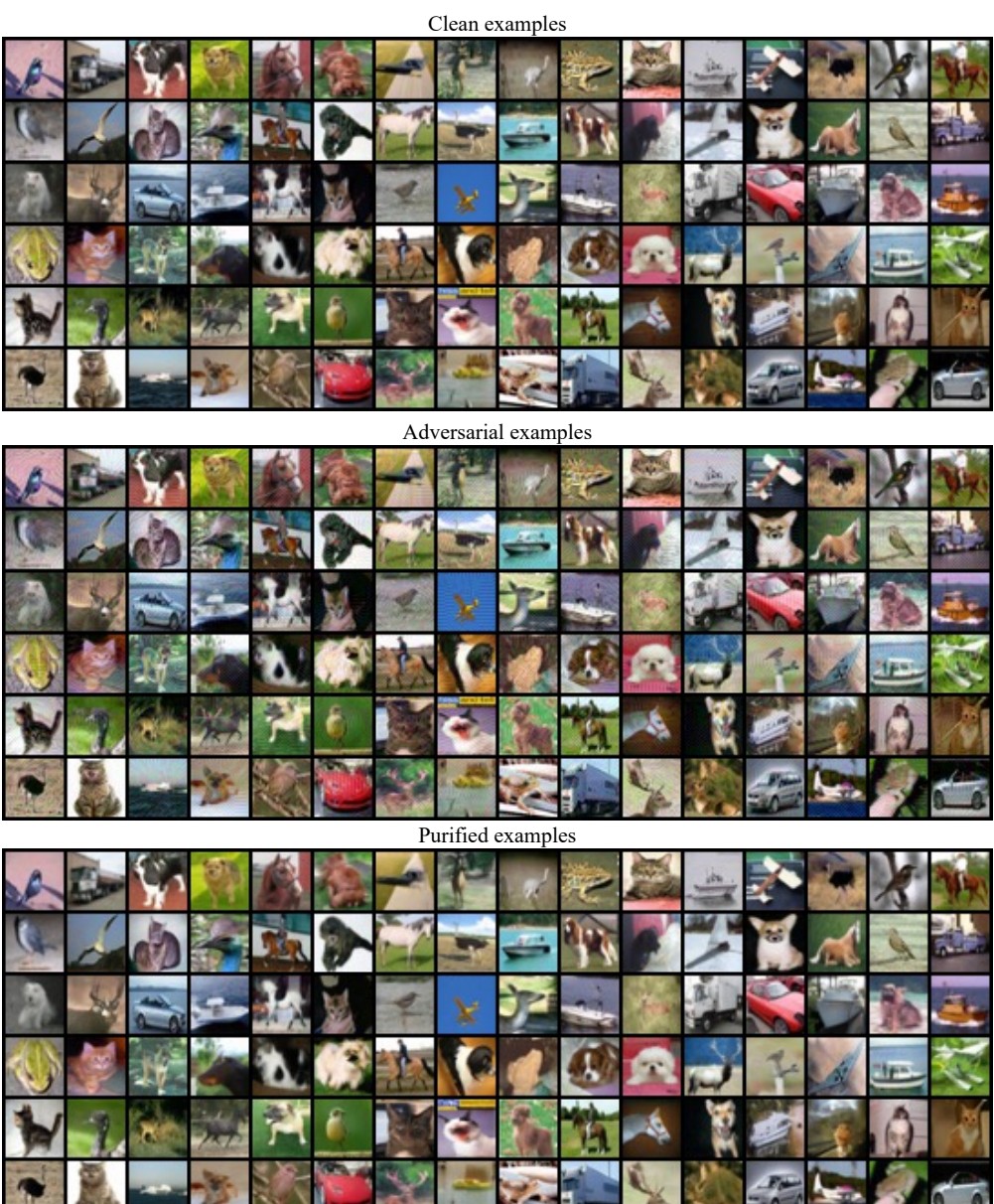

Figure 4: Clean examples (Top), adversarial examples (Middle) and purified examples (Bottom) of CIFAR-10.

Clean examples

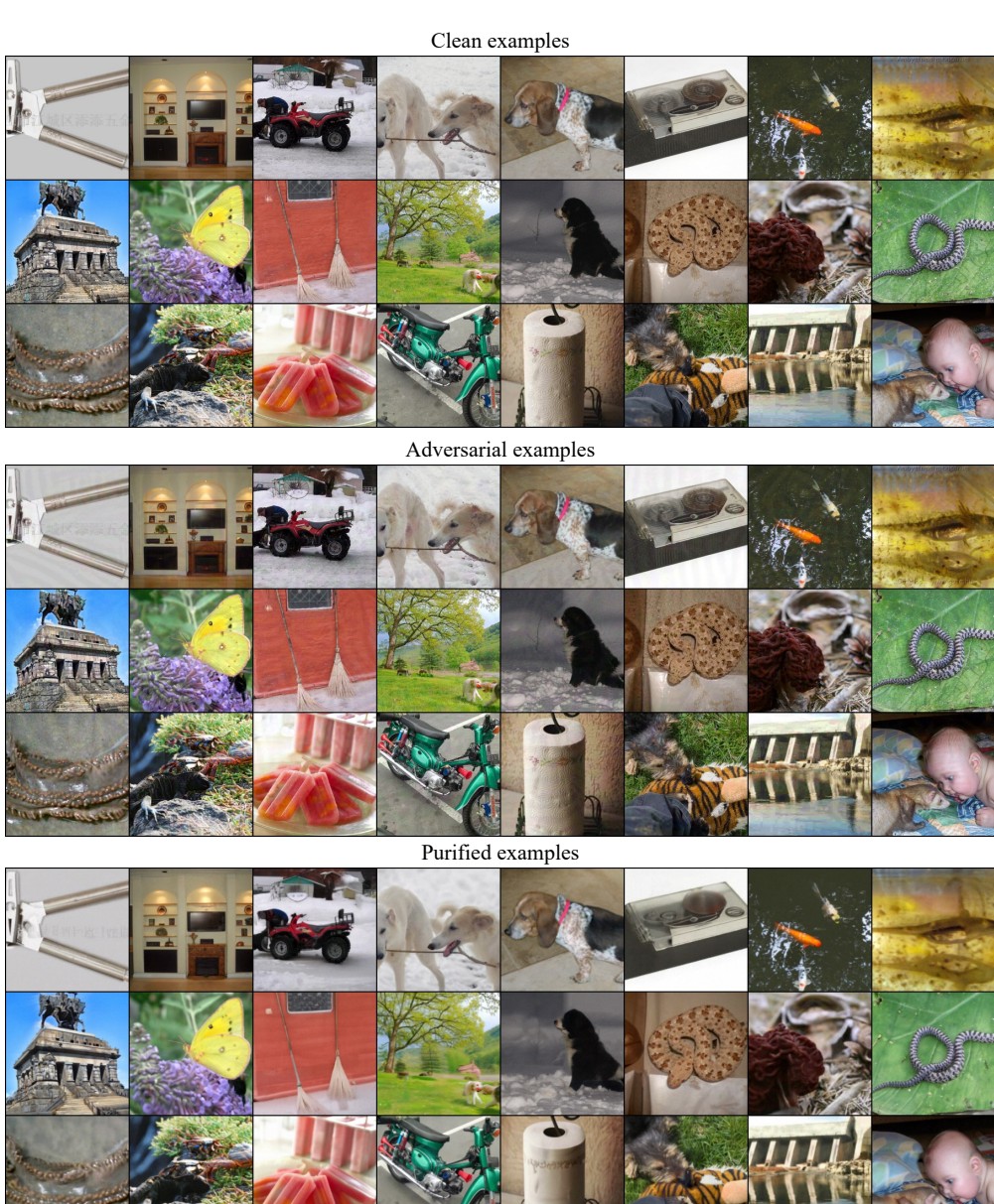

Figure 5: Clean examples (Top), adversarial examples (Middle) and purified examples (Bottom) of ImageNet.

