# OpenReview forum: "Adversarial Guided Diffusion Models for Adversarial Purification"
_ICLR.cc/2025/Conference — ICLR 2025 Conference Withdrawn Submission_

### Official Review · Reviewer_xPEV · 2024-10-27

**Soundness:** 2
**Presentation:** 3
**Contribution:** 2
**Rating:** 5
**Confidence:** 5

**Summary:**

This paper proposed a diffusion model-based adversarial purification (AP) method called adversarial guided diffusion models (AGDM). AP methods face a trade-off between robust and standard accuracy since we have to drop some label semantics to preserve from recovering the adversarial perturbation during the reverse process.

To overcome this, training-free conditional-based diffusion has been introduced into AP, where we can measure the distance between the intermediate results and the known adversarial samples to keep the label semantic. However, the trade-off still exists since the distance metric will recover the adversarial perturbation again.

AGDM solves this challenge by introducing an auxiliary neural network pre-trained via an adversarial training paradigm. This auxiliary neural network could be regarded as a feature extractor that maps the images into the latent space. AGDM further argues that the adversarial training paradigm will help auxiliary neural networks extract the latent that does not suffer from the influence of adversarial perturbation. In this condition, we could use the clean latent feature as the condition to guide the generation process, thus alleviating the trade-off. Meanwhile, auxiliary neural networks could offer logit to enrich the condition, further enhancing the AP.

**Strengths:**

1. The overall method is technically sound. AGDM proposes to train an auxiliary classifier via adversarial training, which is sound. Adversarial training could help the classifier recognize the adversarial sample, which could naturally generate the most robust latent features to defend against the adversarial perturbation. Leveraging this, the overall conditional generation process will be more robust against the adversarial perturbation and alleviate the trade-off.

2. Introducing adversarial training to enhance AP is interesting.

3. The experiments show that AGDM achieves the SOTA performance.

**Weaknesses:**

1. The contributions of this paper are limited. One of the main contributions of this paper is how to calculate the adversarial guidance, which is well explored under the training-free conditional diffusion such as FreeDom. In the view of the FreeDom, it is just a multi-conditional guidance, which could be easy to calculate.

Specifically, the adversarial guidance in Sec. Methods contains two parts: 1) The MSE in the latent space between the intermediate results $x_{t}$ and the adversarial samples$x^{adv}$. 2) A logit of $p_{\phi}(x_{t})$, where $p_{\phi}$ is the classifier from auxiliary neural networks. This could be conducted in the FreeDom as $\nabla_{x_{t}}\log p ({c^{1},c^{2}}|x_{t})$.
Then,
$\nabla_{x_{t}}\log p ({c^{1},c^{2}}|x_{t}) = \nabla_{x_{t}}\log p ({c^{1}}|x_{t})  + \nabla_{x_{t}}\log p ({c^{2}}|x_{t}) $, where
$ \nabla_{x_{t}}\log p ({c^{1}}|x_{t}) =p_{\phi}(x_{t}) $ and $\nabla_{x_{t}}\log p ({c^{2}}|x_{t}) = D(c_{\phi}(x_{t}) - c_{\phi}(x^{adv}))$. $c_{\phi}$ is the feature extracted from the auxiliary neural networks given the parameter $\phi$. In this condition, we could achieve the conditional generation based on $\nabla_{x_{t}}\log p ({c^{1},c^{2}}|x_{t})$ i,e, the adversarial guidance. This process is even simpler than Sec. Methods, which weakens one of the main contributions of this paper.

The other main contribution of this paper is introducing an auxiliary classifier via adversarial training. However, it directly leverages the TRADES method without any new insight. This raises a new question: Does AGDM really alleviate the trade-off for AP? The advantage of the AP compared to the adversarial training is to defend against unseen attacks. After introducing adversarial training (AT), how to ensure that AGDM can defend against unseen attacks? The weakness of AT is that it is difficult to defend against unseen attacks since there are no training samples from unseen attacks in AT. In this condition, AGDM generates a new trade-off between AP and AT again.

To sum up, the contribution of this paper seems to verify that an auxiliary classifier via AT could enhance the AP in some conditions, which seems limited.

2. The experiments are not enough to prove the superiority of the AGDM. (1). It lacks the BPDA [2] attacks. BPDA is one of the important attacks to test the performance of AP, which should be discussed. (2) The performance of AGDM is fair. ZeroPur [3] reports the robust accuracy of AutoAttack on CIFAR-10 ($\epsilon=8/255$) is 82.76%, better than the 78.12% reported in AGDM. Meanwhile, the AP for ZeroPur even drops the diffusion models. AGDM introduces the unconditional diffusion models and an auxiliary classifier, which should be better than the ZeroPur, since there is richer previous knowledge for AGDM.

[1] FreeDoM: Training-Free Energy-Guided Conditional Diffusion Model. Yu, Jiwen and Wang, Yinhuai and Zhao, Chen and Ghanem, Bernard and Zhang, Jian. ICCV.

[2] Diffusion Models for Adversarial Purification. Nie, Weili and Guo, Brandon and Huang, Yujia and Xiao, Chaowei and Vahdat, Arash and Anandkumar, Anima. ICML

[3] ZeroPur: Succinct Training-Free Adversarial Purification. Xiuli Bi and Zonglin Yang and Bo Liu and Xiaodong Cun and Chi-Man Pun and Pietro Lio and Bin Xiao. ArXiv.

**Questions:**

1. How does different AT influence the performance of AGDM? The motivation for this question is from the Weaknesses. 1. AT will inevitably introduce additional problems. For example, if we use PGD to generate training samples for the auxiliary classifier via AT, the classifier's performance will be worse when facing C\&W attack. Is there a way to make the auxiliary classifier better defend against unseen attacks?

2. How does the influence of $s$. For example, could we use the different weights for the MSE and the logit $p_{\phi}(x_{t})$, or could we increase $s$? The motivation for this question is that the training-free conditional generation method relies on the setting of the $s$. Thus, we have to discuss its influence.

To sum up, my main concerns are listed in Sec. Weaknesses and Sec. Questions. The overall contribution of this paper seems limited, and the experimental is not enough. Considering introducing the AT to enhance the AP is an interesting topic, I rate it as "marginally below the acceptance threshold".

---

> ### Author Response · Authors · 2024-11-15
>
> We greatly appreciate the comments of the Reviewer xPEV. Below are our responses to the questions raised.
>
> > Q1: The contributions of this paper are limited. The guidance technique is well explored under the training-free conditional diffusion such as FreeDom.
>
> A1: Thank you for sharing an interesting paper. We think there are some significant differences between our work and [1]. Firstly, the problems we focus on are different: The adversarial perturbations are carefully designed, and their harmful impact on the generated images far exceeds the shortcomings of the generative model itself. When we use training-free conditional diffusion for standard classification tasks, it indeed achieves better performance, but it does not learn robust representations, resulting in poor performance on robust classification tasks. Secondly, due to the difference in problems, there are differences in our algorithms as well. As you mentioned, we can directly minimize the distance between adversarial examples and clean examples as one of the multi-conditional guidance. Wang et al. [2] propose a similar guidance following this idea, but they find that in the training-free mode when the distance between the adversarial and clean examples is too small, it is easy to preserve the perturbations, which is also highlighted and discussed in our paper. Therefore, we propose a new adversarial guidance for robust classification tasks. Although there is a similarity in the high-level concepts, the problems and specific algorithms are different.
> [1] Yu et al. FreeDoM: Training-Free Energy-Guided Conditional Diffusion Model. ICCV, 2023.
> [2] Wang et al. Guided Diffusion Model for Adversarial Purification. Arxiv, 2022.
>
> > Q2: Does AGDM really alleviate the trade-off for AP?
>
> A2: AGDM can really defend against unseen attacks. Firstly, we would like to explain why AP can defend against unseen attacks, but AT cannot. From the pipelines, the answer is straightforward. AT, by 'overfitting' to specific adversarial examples, enables the classifier to effectively defend against those attacks, which obviously cannot defend against unseen attacks. Pre-trained generator-based AP methods first utilize the random transforms to destroy the perturbations, followed by a denoising operation using a generative model. Therefore, once the perturbations can be effectively destroyed, the entire system can defend against all attacks. There is more discussion in section 3.1, and the experiments support that AGDM can defend against unseen attacks in Tables 5, and 8.

---

> ### Author Response · Authors · 2024-11-15
>
> > Q3: The experiments are not enough: It lacks the BPDA attacks. The performance of AGDM is not better than ZeroPur [3], which even drops the diffusion models.
>
> A3:
> 3.1: In evaluating diffusion-based AP method, considering the robustness misestimation caused by obfuscated gradients of the purifier model, the methods should not be evaluated by non-adaptive attacks. Nie, Lee, and Lin et al. [4,5,6] have all discussed this issue in the paper and have employed adaptive attacks to evaluate the methods. On the other hand, although BPDA+EOT is used as a robust evaluation of AP methods in Nie et al., in the latest research, Lee \& Kim find that BPDA+EOT is unsuitable for diffusion-based AP methods tasks. The table below compares the robust accuracy of BPDA+EOT with the lowest robust accuracy between PGD+EOT and AutoAttack. The evaluation results of BPDA+EOT are far less reliable than those of PGD+EOT / AutoAttack. Therefore, the latest evaluation paradigm has been established with AutoAttack, StAdv, and PGD+EOT. We strictly adhere to the established pipeline to evaluate our experiments and follow the same settings to maintain consistency, ensuring the reliability of our results.
>
> | Robust accuracy (%) | | |
> |:--------:|:----------:|:-----------:|
> | Diffusion-based AP   | BPDA+EOT | PGD+EOT / AutoAttack |
> | DISCO                 | 47.18             | 0.00     |
> | ADP                   | 66.91             | 33.48    |
> | GDMP                  | 75.59             | 24.06    |
> | DiffPure              | 81.45             | 46.84    |
> | Lee & Kim (2023)      | -                 | 55.82    |
> | Ours                  | -                 | 64.06    |
>
> [3] Bi et al. ZeroPur: Succinct Training-Free Adversarial Purification. ArXiv, 2024.
> [4] Nie et al. Diffusion Models for Adversarial Purification. ICML, 2022.
> [5] Lee and Kim. Robust Evaluation of Diffusion-based Adversarial Purification. ICCV, 2023.
> [6] Lin et al. Adversarial Training on Purification (AToP): Advancing Both Robustness and Generalization. ICLR, 2024.
>
> 3.2: Focusing on the community surrounding this paper, which aims to improve the robustness of the pre-trained generative model for adversarial purification, our method (AGDM) is highly efficient compared to existing defense methods.
>
> Although ZeroPur can achieve better performance even drops the diffusion models, we believe our work is still meaningful. As described by Wang et al., with the continuous development of diffusion models, using better DMs can achieve superior performance [7]. In the future, the latest DMs are likely to surpass the performance of ZeroPur again, and our method will further optimize these DMs to achieve state-of-the-art results. At the same time, as ICLR (https://iclr.cc/Conferences/2025/ReviewerGuide) claims that: A lack of state-of-the-art results does not by itself constitute grounds for rejection. Submissions bring value to the ICLR community when they convincingly demonstrate new, relevant, impactful knowledge.
>
> [7] Wang et al. Better Diffusion Models Further Improve Adversarial Training. ICML, 2023.
>
> > Q4: Is there a way to make the auxiliary classifier better defend against unseen attacks?
>
> A4: As described in A2, the generalization ability to defend against unseen attacks primarily comes from random transforms, not the auxiliary network.
>
> > Q5: How does the influence of $s$? Could we use the different weights for the MSE and the logit?
>
> A5: We have conducted this experiment in Figure 3c. When we use different weights, we can adjust the balance between standard accuracy and robust accuracy.

---

### Official Review · Reviewer_QfvL · 2024-11-03

**Soundness:** 2
**Presentation:** 3
**Contribution:** 2
**Rating:** 3
**Confidence:** 4

**Summary:**

This paper introduces an improved diffusion-based adversarial purification (AP) method termed adversarial guided diffusion model (AGDM). To address the limitation of existing diffusion-based AP methods that the lack of proper guidance can lead to shifting towards incorrect classes, AGDM utilizes a robust auxiliary neural network obtained by TRADES to guide the diffusion model in AP. Experimental results suggest that AGDM can improve the robust accuracy compared with the diffusion-based AP baselines.

**Strengths:**

- The limitations of existing related methods are thoroughly discussed, and the proposed AGDM is well-motivated.
- The superiority of AGDM to existing diffusion-based AP methods indicates the significance of guidance for the diffusion model in AP.

**Weaknesses:**

- The notations and interpretations in Section 3.2 can be confusing. Specifically, the interpretation of $p_{\phi}(x' \mid x_t)$ (Lines 212-213) only concerns the semantic information of $x'$, but the notation itself seems to indicate that the specific pixel values of $x'$ are also concerned. If only the semantic information is considered, it should be something like $p_{\phi}(s(x') \mid x_t)$.
- In Lines 277-278, it is stated that the auxiliary network is not required to be a robust classifier but to have adversarially robust representations. From my perspective, the difference between the two requirements is that the latter does not consider the classification ability of the network (e.g., it can be a self-supervised model), but it seems that a non-classification model may not suffice for the guidance of AGDM.
- The architecture of the auxiliary network (or whether it is the same as the classifier) is not stated in the experimental settings.
- The number of PGD and EOT steps is not stated. Insufficient PGD and EOT iterations can lead to the overestimation of the robustness of AP methods.
- Judging from Figure 3(a), the robust accuracy of the AGDM is not significantly higher than that of AT methods under $\ell_{\infty}$ attacks. It might be the case that using state-of-the-art AT models as the auxiliary network may further improve the performance of AGDM, but no evidence is provided.
- The practicality of the proposed AGDM may be questionable. High computation costs are required for both training (AT of the auxiliary network) and inference (iterative denoising process for purification). It is argued that the adversarial fine-tuning of DMs in AToP is computationally expensive (Lines 079-080), but the proposed method also suffers from the same issue. There is also a lack of evidence for the transferability of a pre-trained AGDM to models of other architectures or for different tasks, which may indicate the practical value of the AP method.

**Questions:**

- In Lines 209-210, why can we assume that $y$ and $x'$ are conditionally independent given $x_t$?

---

> ### Author Response · Authors · 2024-11-15
>
> We greatly appreciate the comments of the Reviewer QfvL. Below are our responses to the questions raised.
>
> > Q1: The notations and interpretations in Section 3.2 can be confusing. $p_\phi(x' \mid x_t)$ should be written as something like $p_\phi(s(x') \mid x_t)$.
>
> A1: The parametric form of $p_\phi(x' \mid x_t)$ used in our model is explicitly defined in Eq. (2). Line 212-213 and Eq. (1) are general descriptions about the guided purification.
>
> > Q2: In Lines 277-278, it is stated that the auxiliary network is not required to be a robust classifier but to have adversarially robust representations. It seems that a non-classification model may not suffice for the guidance of AGDM.
>
> A2: Thanks for the sharp insights and let us explain further. With the development of AI, various non-classification tasks have emerged and more research has begun to discuss whether only studies on robustness in classification tasks are limited. We believe that AP is a promising technology that is not limited to classification. Here, we want to clarify that the auxiliary network is also not limited to classification. Of course, if used for classification tasks, a classifier-based model is required; for other tasks, different models are required. The constant factor is that AP needs to provide robust representations or data. But we agree with you that this sentence may be an over-claim in this paper, and we will revise it accordingly.
>
> > Q3: Is the architecture of the auxiliary network the same as the classifier?
>
> A3: The architecture of the auxiliary network is not required to be consistent with the classifier. All results use the WRN28-10 architecture to train the auxiliary network. In the paper, we evaluate it on various classifiers (ResNet-50, WRN28-10, WRN70-16), and all achieve good performance.
>
> > Q4: What are the number of PGD and EOT steps?
>
> A4: The EOT iteration is 20 and PGD step is 200, the same settings in [1,2].
> [1] Nie et al. Diffusion Models for Adversarial Purification. ICML, 2022.
> [2] Lee and Kim. Robust Evaluation of Diffusion-based Adversarial Purification. ICCV, 2023.
>
> > Q5. How does the performance of different architectures as the auxiliary neural network?
>
> A5: We tried the ResNet-18 and WideResNet-28-10 architectures in the early stages of experimentation. WRN-28-10 performed better, but there was no significant difference compared to ResNet-18. However, this is not the main goal that the paper aims to solve, so we did not conduct an in-depth study of these experiments. We will consider adding this part of the experiment to the supplementary materials.
>
> > Q6: The proposed method may be questionable. Both the training and the inference are time-consuming. There is also a lack of evidence for the transferability for different architectures or tasks.
>
> A6:
> 6.1: Focusing on the community surrounding this paper, which aims to improve the robustness of the pre-trained generative model for adversarial purification, our method (AGDM) is highly efficient compared to existing defense methods.
> The additional training is for the auxiliary neural network (ANN), which only consumes a small cost. Within one A5000 GPU, training the ANN on CIFAR-10 takes only ~3.17 hours, which is an entirely acceptable training cost. Compared to pre-trained diffusion-based AP, we indeed increase the cost of training since the fine-tuning cost for the former is 0. However, as Lin et al. [3] proposed AToP and have pointed out that additional training is necessary to improve the robustness of the current pre-trained generator-based AP. Furthermore, due to the significant training costs, they claim that AToP cannot work on the DM-based AP. In contrast, we only need to train a small neural network, which significantly reduces the training costs compared to AToP. Moreover, it is worth noting that while our method significantly improves performance, it has almost the same inference time as the vanilla diffusion-based AP method, as shown in the table below.
>
> |Training on CIFAR-10 |with 1000 images|||
> |:-:|:-:|:-:|:-:|
> | Methods | AToP on GAN | AToP on DM | Ours|
> |Time| ~62 s| ~144 min| ~17 s|
>
> | Inference time (s) | | | |
> |:-:|:-:|:-:|:-:|
> | Methods   | CIFAR-10 | CIFAR-100 | ImageNet |
> | DiffPure  |1.49| 1.50| 5.11|
> | Ours  | 1.73 | 1.75| 5.52|
>
> [3] Lin et al. Adversarial Training on Purification (AToP): Advancing Both Robustness and Generalization. ICLR, 2024.
>
> 6.2: We respectfully point out that we have conducted extensive experiments on multiple tasks (CIFAR-10, CIFAR-100, ImageNet) across various attacks (PGD, EOT, AutoAttack and StAdv), architectures(ResNet-50, WRN28-10, WRN70-16), norms($l_2$, $l_\infty$, non-$l_p$) to demonstrate the strong generalization ability of our method.
>
> > Q7: In Lines 209-210, why can we assume that $y$ and $x'$ are conditionally independent given $x_t$?
>
> A7: Since $x_t$ contains information about $y$ and $x'$, we assume they are conditionally independent given $x_t$.

---

### Official Review · Reviewer_bRNc · 2024-11-03

**Soundness:** 3
**Presentation:** 2
**Contribution:** 1
**Rating:** 3
**Confidence:** 4

**Summary:**

This work proposes an adversarial purification method for defending against adversarial examples. The proposed model uses a robust classifier to guide the reverse process of DMs during purification, helping to preserve semantic information and improve robustness against adversarial perturbations. Experiments conducted on several datasets demonstrated the effectiveness.

**Strengths:**

1. The proposed method seems to be reasonable. Combining the robust classifier with diffusion models has the potential to improve the robustness.

2. The experiments are relatively comprehensive, with several datasets and several attack methods included.

**Weaknesses:**

1. The tricky illustrations. The diffusion step t is set to be 70 in the experiments, while in Fig 1, the step is set to 400. I recommend using the actual step for illustration to help readers comprehensively understand this work. Furthermore, How to create Figure 2 is not clear and there is no experimental support for Figure 2.
2. There is no theoretical analysis to show the reason why this process is better than other classifier-guided diffusion purification methods.
3. Lack of innovation and contribution. The core contribution of this work is to replace the standard classifier with a robust classifier in the framework of guided diffusion models [1], which do not bring a new perspective on diffusion-based purification. Therefore, the innovation is questionable.
4. The results  are not convincing. According to [2], the robustness of diffusion-based purification is significantly over-estimated and the robustness should be reported under adaptive attacks. I recommend all results in this work be reported following the settings in [2].
5. The chaotic formulations. Please follow the symbolics in DDPM or Score-SDE. For example, $\mathbf{x}$ instead of $x$ should be used to represent an image.

[1] Guided diffusion model for adversarial purification

[2] Robust Classification via a Single Diffusion Model

**Questions:**

Please see the weaknesses above.

---

> ### Author Response · Authors · 2024-11-15
>
> We greatly appreciate the comments of the Reviewer bRNc. Below are our responses to the questions raised.
>
> > Q1: The illustrations are not clear and there is no experimental support.
>
> A1: For Figure 1, if only 70 steps are used, the modifications to the input image are minimal, and there is no significant difference in the output purified images. Therefore, in order to make a clearer comparison between our method and existing work, we set a larger step in this conceptual diagram. For Figure 2, we respectfully point out that we have extensively described it in Section 3.1 and provided the experiments to support this idea in Tables 4, and 6.
>
> > Q2: There is no theoretical analysis to support the proposed method.
>
> A2: We respectfully point out that according to the reviewer guidelines provided by ICLR (https://iclr.cc/Conferences/2025/ReviewerGuide), the requirement of 'Does the paper support the claims?' is that 'This includes determining if results, **whether theoretical or empirical**, are correct and if they are scientifically rigorous.' Given our paper is not a theoretically oriented paper, we believe empirical evaluation is acceptable, and lack of theoretical analysis cannot be the reason for rejecting the paper.
>
> > Q3: Lack of innovation and contribution, which do not bring a new perspective on diffusion-based purification.
>
> A3: Focusing on the community surrounding this paper, which aims to improve the robustness of the pre-trained generative model for adversarial purification, our method (AGDM) is highly efficient and completely different from the existing DM-based AP methods. In the original manuscripts, we have extensively discussed the paper [1] you provided, Wang et al. [1] proposed a training-free conditional DM that may result in preserving perturbations. To address this issue, we introduced a trainable guidance, which is completely a new perspective compared with [1]. In summary, 1. We propose a new robust guidance to optimize the reverse process to generate robust purified examples. 2. We are the first to consider the accuracy-robustness trade-off in pre-trained DM-based AP, which might be a significant contribution to advance the development of this field. 3. We conduct the experiments on two most important evaluation methods and recommend that we should consider the worst-case in evaluation of DM-based AP.
> [1] Wang et al. Guided Diffusion Model for Adversarial Purification. Arxiv, 2022.
>
> > Q4: The results are not convincing. I recommend all results in this work be reported following the settings in [2].
>
> A4: We strongly disagree with your claim that our results are not convincing, and your view that all results should follow the settings in [2].
> 1. Chen et al. [2] claim that the robustness is over-estimated when evaluating the diffusion-based model on AutoAttack, and share the same idea with Lee \& Kim [3], using PGD+EOT to evaluate the diffusion classifier. We have cited these papers [2, 3] and conducted the experiments on both PGD+EOT and AutoAttack, using the worst-case robust accuracy as a more reliable evaluation.
> 2. Furthermore, Chen et al. [2] focus on classifier architecture, and we focus on adversarial purification defense methods. Our paper and [2] address completely different tasks of robustness, whereas the paper [3] that we follow aligns exactly with our task. Although many evaluation methods are universal, it is unreasonable to insist that we should not follow an AP evaluation paper [3] and forcefully recommend that we follow [2].
> 3. As you mentioned, evaluating AP methods only using AutoAttack would lead to overestimating the robustness, however, evaluating AT methods using PGD+EOT would also lead to overestimating, as shown in Figure 3. Therefore, we conduct the experiments using PGD+EOT and AutoAttack, regarding the worst-case robust accuracy as the metric for assessing robustness of methods.
> [2] Chen et al. Robust Classification via a Single Diffusion Model. ICML 2024.
> [3] Lee and Kim. Robust Evaluation of Diffusion-based Adversarial Purification. ICCV, 2023.
>
> > Q5: The chaotic formulations. Please follow the symbolics in DDPM or Score-SDE. For example, $\textbf{x}$ instead of $x$ should be used to represent an image.
>
> A5: We strongly disagree with this weakness. For an independent paper, we need to ensure that within this paper, all symbols are self-contained, not necessarily the same as those in another paper. Also, if I maintain the same symbolic notation as in DDPM, there may be another reviewer who says: Please follow the symbols in DM-based AP, using $x$ instead of $\textbf{x}$. We believe this weakness is baseless.

---

### Official Review · Reviewer_MYy2 · 2024-11-04

**Soundness:** 2
**Presentation:** 3
**Contribution:** 2
**Rating:** 3
**Confidence:** 4

**Summary:**

Existing DM-based AP methods have no explicit guidance or improper guidance (e.g., existing guidances may also preserve adversarial perturbations in purified examples). To solve this issue, this paper proposes AGDM, which adversarially trains an auxiliary neural network to provide more robust guidance in the reverse process of DMs. Experiments show that AGDM can improve robustness by a notable margin.

**Strengths:**

1. The motivation of the paper is very clear, which effectively delivers the main insight of this paper.

2. The proposed method is intuitive and easy to understand.

3. The guided sampling can be extended to continuous-time DMs,  which means AGDM can be generalized to different DMs.

**Weaknesses:**

1. The proposed method may have very low efficiency. DM-based AP method is very slow during the inference stage (as it is a completely 'inference-time defense'). Based on this, this paper proposes to train an auxiliary neural network using AT, which will further increase the computational complexity of both the training (as AT is very slow by its nature) and the inference (as adversarial guidance is introduced to the reverse process of DMs).

2. A followed-up weakness is: this paper did not report the training time for the auxiliary neural network and the inference time for the entire defense. Therefore, it is unclear whether the improvement obtained by AGDM is worthwhile in terms of the sacrifice in efficiency. I hope the authors could clarify this during the rebuttal.

3. The experiment settings are very unclear. Firstly, what are the seed numbers for those 512 images during the evaluation, are they consistent with DiffPure (as this paper said following DiffPure in line 327)? Secondly, what are the iteration numbers for PGD+EOT and AutoAttack, are they the same across all the baseline methods? Thirdly, it is unclear whether the adversarial examples for evaluations are generated under a white-box setting (i.e., attack AGDM + classifier as a whole) or a grey-box setting (i.e., only attack the classifier, or only attack the vanilla DMs). A fair evaluation process is very important in this field, so I would strongly encourage the authors to include detailed experiment settings during the rebuttal.

4. This paper lacks of ablation studies on the auxiliary neural network. What's the architecture of the auxiliary neural network? How the performance would be affected if other architectures are used?

**Questions:**

Most questions that I hope the authors can address are given in the weaknesses, and here are some additional questions:

1. Just for curiosity, given that DM-based AP methods and AT methods are completely different, what's the motivation for comparing DM-based AP methods with AT methods here? I can see most experiments follow what was done in DiffPure (except the PGD+EOT experiments), but DiffPure compared to AT methods because it is the first DM-based AP method and thus it cannot find another DM-based AP method to compare with. However, after a few years, now there are many DM-based AP methods in this field and I think comparing to DM-based AP methods is enough to demonstrate the effectiveness of AGDM. I hope authors could share their ideas about this question during the rebuttal.

2. A follow-up question is: if it is necessary to compare with AT methods, what do you think is the most fair way to do so? According to [1], AT methods perform worse on AutoAttack while DM-based AP methods perform worse on PGD+EOT. So a natural question is: how can we compare DM-based AP methods with AT methods in a fair setting?

[1] Minjong Lee and Dongwoo Kim. Robust evaluation of diffusion-based adversarial purification. In ICCV 2023.

I am willing to increase my rating if the authors can address my concerns during the rebuttal. Also, if I misunderstood any part of the paper, feel free to correct me.

---

> ### Author Response · Authors · 2024-11-15
>
> We greatly appreciate the comments of the Reviewer MYy2. Below are our responses to the questions raised.
>
> > Q1: The proposed method may have very low efficiency. Both the training and the inference are time-consuming.
>
> A1: Focusing on the community surrounding this paper, which aims to improve the robustness of the pre-trained generative model for adversarial purification, our method (AGDM) is highly efficient compared to existing defense methods.
> The additional training is for the auxiliary neural network (ANN), which only consumes a small cost. Within one A5000 GPU, training the ANN on CIFAR-10 takes only ~3.17 hours, which is an entirely acceptable training cost. Compared to pre-trained diffusion-based AP, we indeed increase the cost of training since the fine-tuning cost for the former is 0. However, as Lin et al. [1] proposed AToP and have pointed out that additional training is necessary to improve the robustness of the current pre-trained generator-based AP. Furthermore, due to the significant training costs, they claim that AToP cannot work on the DM-based AP. In contrast, we only need to train a small neural network, which significantly reduces the training costs compared to AToP. Moreover, it is worth noting that while our method significantly improves performance, it has almost the same inference time as the vanilla diffusion-based AP method, as shown in the table below.
>
> | Training on CIFAR-10 | with 1000 images | | |
> |:--------:|:----------:|:-----------:|:----------:|
> | Methods   | AToP on GAN | AToP on DM | Ours |
> | Time  | ~62 s | ~144 min | ~17 s |
>
> | Inference time (s) | | | |
> |:--------:|:----------:|:-----------:|:----------:|
> | Methods   | CIFAR-10 | CIFAR-100 | ImageNet |
> | DiffPure  | 1.49 s    | 1.50 s     | 5.11 s    |
> | Ours      | 1.73 s    | 1.75 s     | 5.52 s    |
>
> [1] Lin et al. Adversarial Training on Purification (AToP): Advancing Both Robustness and Generalization. ICLR, 2024.
>
> > Q2: The paper did not report the training time and the inference time.
>
> A2: We conduct experiments on training and testing times in the above table. In summary, compared to existing methods, our method can save a significant amount of training costs and has similar inference costs.
>
> > Q3: What is the seed during training? What are the iteration numbers for PGD+EOT and AutoAttack? What is the object of the attack?
>
> A3: All experiments in this paper keep the same settings in [2,3]. The code will be available upon acceptance to clearly show details.
> 3.1: The seed is 777, a random lucky number. In the ablation experiments (Table 4), we used the same seed to ensure reliability.
> 3.2: The EOT iteration is 20, the PGD step is 200, and the AutoAttack version is rand, the same settings in [2,3].
> 3.3: All experiments are conducted on attacking the whole process, i.e., Attack{classifier(AGDM(x))}.
>
> [2] Nie et al. Diffusion Models for Adversarial Purification. ICML, 2022.
> [3] Lee and Kim. Robust Evaluation of Diffusion-based Adversarial Purification. ICCV, 2023.
>
> > Q4: The paper lacks ablation studies for the auxiliary network. How does the performance of different architectures as the auxiliary neural network?
>
> A4:
> 4.1: We have conducted ablation experiments in Table 4.
> 4.2: We tried the ResNet-18 and WideResNet-28-10 architectures. WRN-28-10 performed better, but there was no significant difference compared to ResNet-18. In the paper, we use WRN-28-10 architecture for all experiments.
>
> > Q5: What's the motivation for comparing DM-based AP methods with AT methods?
>
> A5: Our ultimate goal is to create a robust system to achieve better robust accuracy, not just to optimize the AP issue solely. The existing studies have shown that AT and AP have their unique advantages, none of them has an absolute advantage across various settings. Therefore, our method represents a high-level integration of AT and AP and it is necessary to compare it with both AT and AP methods.
>
> > Q6: What do you think is the most fair way to do so?
>
> A6: It's a good question that we also mentioned in the paper. However, this paper does not delve into the evaluation method, so we cannot provide a definitive answer, we can only offer a preliminary idea is that: we should consider the worst-case results of the robust accuracy across PGD+EOT and AutoAttack as shown in Figure 3.

---

### Note · Authors · 2024-11-15

I have read and agree with the venue's withdrawal policy on behalf of myself and my co-authors.